# Revive Re-weighting in Imbalanced Learning by Density Ratio Estimation

**Jiaan Luo**[1,3†]   **Feng Hong**[1†]   **Jiangchao Yao**[1,3‡]   **Bo Han**[4]   **Ya Zhang**[2,3]   **Yanfeng Wang**[2,3]

[1]Cooperative Medianet Innovation Center, Shanghai Jiao Tong University
[2]School of Artificial Intelligence, Shanghai Jiao Tong University
[3]Shanghai Artificial Intelligence Laboratory
[4]Hong Kong Baptist University
`{luojiaan, feng.hong, Sunarker, ya_zhang, wangyanfeng}@sjtu.edu.cn`
`bhanml@comp.hkbu.edu.hk`

## Abstract

In deep learning, model performance often deteriorates when trained on highly imbalanced datasets, especially when evaluation metrics require robust generalization across underrepresented classes. To address the challenges posed by imbalanced data distributions, this study introduces a novel method utilizing density ratio estimation for dynamic class weight adjustment, termed as Re-weighting with Density Ratio (RDR). Our method adaptively adjusts the importance of each class during training, mitigates overfitting on dominant classes and enhances model adaptability across diverse datasets. Extensive experiments conducted on various large scale benchmark datasets validate the effectiveness of our method. Results demonstrate substantial improvements in generalization capabilities, particularly under severely imbalanced conditions. The code is available here.

## 1 Introduction

In recent years, deep learning has made significant strides across various domains by utilizing complex architectures and large-scale datasets, setting new benchmarks for performance. However, these advancements often rely on well-curated datasets that ensure balanced class distributions [Russakovsky et al., 2015]. In contrast, real-world datasets typically exhibit a long-tailed distribution, where few classes dominate the majority of samples, while many others are underrepresented [Krizhevsky et al., 2009]. This imbalance leads to model biases favoring frequent classes, thereby reducing performance on the less common ones. Yet, in many applications—such as medical diagnostics and financial analysis—greater emphasis is placed on ensuring strong generalization for underrepresented classes. Addressing this challenge not only reduces data collection costs but also improves the robustness and fairness of the models.

Many excellent methods, such as re-sampling [Bowyer et al., 2011], re-weighting [Morik et al., 1999], decoupled learning [Kang et al., 2020], margin-based learning [Cao et al., 2019, Menon et al., 2020], transfer learning [Yin et al., 2019] and contrastive learning [Tian et al., 2021], have been proposed to tackle the issue of imbalanced data. Despite the simplicity of re-weighting, it falls behind in performance significantly compared with other directions of methods due to the inappropriate weighting coefficients during training. Cui et al. [2019] proposes a method for re-weighting by effective number, which accounts for potential overlaps among data samples and adjusts the weights for each category based on the actual effective number of samples. Chen et al. [2023b] leverages the effective area to re-weight, considering the actual spanned space of each class. However, such subsequent improvements can alleviate but still cannot effectively push that forward. Wang et al.

---

[†]The first two authors contribute equally.

[‡]The corresponding author is Jiangchao Yao (`Sunarker@sjtu.edu.cn`).

38th Conference on Neural Information Processing Systems (NeurIPS 2024).

[2023] obtains a fine-grained generalization bound for re-weighting in imbalanced learning through the data-dependent contraction technique. Limited research has focused on the intrinsic limitations of the commonly employed re-weighting-based loss functions and the corresponding balancing mechanisms designed to enhance parity in class representation.

This study rethinks the characteristics of re-weighted loss and explores the question *"Why is re-weighting necessary under conditions of sample imbalance?"* Under conditions of sample imbalance, the variation in weights of samples arises due to discrepancies between the distribution of collected data and a balanced data distribution. In scenarios where class balance exists, such discrepancies are absent, thus obviating the need for re-weighting. Conversely, in imbalanced settings, re-weighting becomes essential to bridge the gap between these distributions. The weights must therefore represent a suitable compromise between balanced and imbalanced distributions and necessarily reflect accurately on each sample. Additionally, as model training progresses dynamically, optimizing the fit to feature distributions, the weights applied to each sample should be continuously updated to maintain robust performance.

This research introduces a novel method, Reweighting with Density Ratio (RDR), designed to mitigate learning disparities in imbalanced distributions. In this method, a feature extractor is employed to discern the features from the training data. A more balanced feature distribution is approximated by continuously updating the momentum on the feature level. This enables real-time density ratio estimation with features learned under imbalanced distributions, thereby obtaining the sample-wise weights. Notably, as the learned features evolve, our method dynamically adjusts weights in response to observed shifts in class density throughout the training cycle, ensuring that the model remains adaptive and effective. This method significantly enhances the robustness and adaptability of the training process. By integrating density ratio estimation to evaluate the difference between the balanced and real data distributions, our approach more accurately reflects the underlying class distribution and improves the model's generalization capabilities across diverse datasets. The contributions are summarized as follows:

- We explore the existing re-weighting techniques, and model the performance of various algorithms during training under different data distributions. This approach offers a novel perspective on understanding re-weighting methods in the scenarios of sample imbalance.

- We introduce a novel methodology, Re-weighting with Density Ratio (RDR), which leverages the method of density ratio estimation to dynamically adjust class weights during model training. This approach not only addresses the limitations of prior re-weighting methods but also introduces a mechanism to continuously adapt to the changing importance of classes as learning progresses, thereby enhancing model robustness and adaptability.

- We conduct extensive experiments to validate the effectiveness of our proposed RDR method. These experiments are conducted across various large-scale, long-tailed datasets, demonstrating substantial improvements in handling class imbalance. Our results illustrate significant enhancements in generalization capabilities, particularly under severely imbalanced scenarios.

## 2 Related Work

### 2.1 Re-weighting Based Methods

Re-weighting methods for addressing class imbalance have evolved significantly over the years. Early techniques, such as [Zadrozny et al., 2003], employed inverse frequency techniques to address class imbalances but failed to consider deeper data distribution traits, leading to sub-optimal outcomes. Addressing these shortcomings, Huang et al. [2016] introduced a cost-sensitive learning framework that, beyond simple frequency adjustments, incorporated misclassification costs to achieve a more nuanced balance. However, this approach still struggled with complexities like class overlap and label noise. To further refine this approach, Lin et al. [2017] developed Focal Loss, which employs a modulation factor based on the prediction probability to adjust the loss function, thereby amplifying the impact of hard-to-classify samples while reducing the loss contribution of easy-to-classify samples. Cui et al. [2019] introduced Class-Balanced Loss, which adjusts loss by data overlap, calculating the effective number of each class. Advancements continued with methods based on training gradients, such as [Ren et al., 2020b, Wang et al., 2021a]. Chen et al. [2023b] proposed Adaptive Re-weighting via effective area, which enhances model accuracy by considering the spatial distribution and density

of data points within classes. Ma et al. [2023] introduced a re-weighting method that adjusts based on semantic richness and visual variability. However, no prior work has tackled the issue of sample imbalance by dynamically re-weighting based on the model's performance across training and test sets with differing distributions during the training process.

## 2.2 Non-re-weighting Based Methods

In addition to re-weighting, many other methods are available to address the issue of sample imbalance. Re-sampling techniques [Kubat and Matwin, 1997, Wallace et al., 2011, Han et al., 2005, Hong et al., 2024a] mitigate category imbalances by under-sampling dominant classes [Buda et al., 2018] or over-sampling minority classes [Bowyer et al., 2011]. However, under-sampling may degrade feature representation by discarding valuable majority class data, whereas over-sampling could cause overfitting by duplicating minority class samples. Decoupled training approaches, such as [Kang et al., 2020], challenge the traditional joint training model by separating representation learning from classification. Margin-based methods such as, LADM [Cao et al., 2019], LA [Menon et al., 2020] and VS [Kini et al., 2021], adjust training processes to increase minority class margins, therefore to obtain a more balanced decision boundary. More flexible and robust methods are proposed, including Transfer Learning [Yin et al., 2019, Liu et al., 2019], Contrastive Learning [Li et al., 2022, Chen et al., 2023a], Ensemble Learning [Wang et al., 2021b, Cai et al., 2021] and Self-supervised Learning [Liu et al., 2022, Zhou et al., 2023b]. Please refer to Appendix B for more discussions.

# 3 Method

## 3.1 Problem Setup

For a typical classification task in imbalanced learning, suppose given a training dataset $\mathcal{S} = \bigcup_{i=1}^{n}\{(\boldsymbol{x}_i, \boldsymbol{y}_i)\}$, where $n$ is the total number of samples. Denote $\{n_1, n_2, ..., n_d\}$ as the sample number of each class. We assume, without loss of generality, that $n_i < n_j$, when $i < j$, with $n_d$ typically much larger than $n_1$, reflecting a pronounced imbalance in class distribution. We use a typical loss function like $l \colon \mathcal{W} \times \mathcal{X} \times \mathcal{Y} \to \mathbb{R}_+$. Denote $\{\pi_1, \pi_2, ..., \pi_d\}$ as the proportions of each class, such that $\sum_{i=1}^{d} \pi_i = 1$. Define a family of deep learning models parameterized by $\omega \in \mathcal{W} \subseteq \mathbb{R}^k$. Typically, a model consists of a feature extractor $f(x; \phi)$ and a classifier $h(z; \theta)$, with $\omega = \bigcup\{\phi, \theta\}$. The notations used in this paper are summarized in Appendix A.

## 3.2 Motivation

Inspired by our review of prior methods, we observe a gap in the adaptation of dynamic class-weight adjustments during training phases. Building on the groundwork of static re-weighting strategies, we introduce a novel approach utilizing the method of density ratio estimation to dynamically recalibrate class weights. This innovation aims to provide a more refined adjustment by estimating real-time class density, thereby promoting an equitable influence of all classes throughout the training.

## 3.3 Dynamic Re-weighting with Density Ratio

In a typical training optimization problem, our objective is to minimize the empirical risk of the loss function, *i.e.*, $\overline{R} = \frac{1}{n} \sum_{i=1}^{n} l(x_i, y_i; \omega)$. However, in imbalanced datasets, where the frequency of samples across different classes varies, it is necessary to adjust for these gaps by applying different weights for the samples. We assume that the weight of each sample is denoted by $\alpha(x, y; \omega)$, then the empirical risk can be formulated like $\overline{R} = \frac{1}{n} \sum_{i=1}^{n} \alpha(x_i, y_i; \omega) l(x_i, y_i; \omega)$.

In naive re-weighting approaches, the weight $\alpha$ of class $y$ is often set to $\frac{1}{\pi_y}$. This setting is based on the assumption that the distribution of the training set $P$ and the distribution balanced data set $P_{bal}$ satisfy the equation $\overline{P}(x|y; \omega) = P_{bal}(x|y; \omega)$. However, in practical training scenarios, both training and test sets are subsets drawn from the actual distribution, leading to potential missing of feature patterns. Furthermore, classes with more complex features and lower sample frequencies tend to exhibit more pronounced missing of patterns. Therefore, in the training process, there exists a discrepancy between $\overline{P}(x|y; \omega)$ and $P_{bal}(x|y; \omega)$. We measure the extent of this discrepancy using the ratio $r(x|y; w) = \overline{P}(x|y; \omega)/P_{bal}(x|y; \omega)$, incorporating it as a correction term into our weighting

scheme. Consequently, the empirical risk can be reformulated as follows

$$\overline{R} = \frac{1}{n}\sum_{i=1}^{n}\frac{r(x_i|y_i;\omega)}{\pi_{y_i}}l(x_i, y_i;\omega) \tag{1}$$

We can explain the rationality of this formula as follows. Considering each class $i$, where $P_{bal}(y_i;\omega) \propto \pi_i^{-1}P(y_i;\omega)$ and $P_{bal}(x_i|y_i;\omega) = r(x_i|y_i;\omega)P(x_i|y_i;\omega)$, with conditional probability formula, we can derive:

$$
\begin{aligned}
R &= \mathbb{E}_P\frac{1}{\pi_i}(r(x_i|y_i;\omega)l(x_i, y_i;\omega)) = \mathbb{E}_P\frac{P_{bal}(y;\omega)}{P(y;\omega)}\frac{P_{bal}(x|y;\omega)}{P(x|y;\omega)}l(x, y;\omega) \\
&= \mathbb{E}_P\left(\frac{P_{bal}(x, y;\omega)}{P(x, y;\omega)}l(x, y;\omega)\right) = \mathbb{E}_{P_{bal}}l(x, y;\omega)
\end{aligned}
\tag{2}
$$

Eq. (2) demonstrates that our approach aligns with the balanced risk of the loss function. Consequently, minimizing Eq. (1) also serves to minimize the balanced risk.

Let's take a closer look at $r(x|y;w) = \overline{P}(x|y;\omega)/P_{bal}(x|y;\omega)$. The variable $r$ represents the ratio of two different distributions. We approximate this ratio using methods of density ratio estimation. This problem can be solved by first-order moment matching approach. Our goal is to minimize

$$\operatorname*{argmin}_r\left\|\int xr(x|y;\omega)P_{bal}(x|y;\omega)\mathrm{d}x - \int xP(x|y;\omega)\mathrm{d}x\right\|^2 \tag{3}$$

where $\|\cdot\|$ denotes the Euclidean norm. Recall that we capture the features of the input samples by the feature extractor $f(x;\phi)$ in our model, and these features are a good reflection of what our model learned from the distribution of the input samples. Therefore, in order to capture more complex structures and patterns in raw data, we use $f(x;\phi)$ to obtain a variant of Eq. (3). Our goal can be achieved by obtaining $\operatorname{argmin}_r \mathrm{MM}'(r)$, where $\mathrm{MM}'(r)$ denotes

$$\left\|\int f(x;\phi)r(x|y;\omega)P_{bal}(x|y;\omega)\mathrm{d}x - \int f(x;\phi)P(x|y;\omega)\mathrm{d}x\right\|^2 \tag{4}$$

where MM stands for 'moment matching'. Let us ignore the irrelevant constant in $\mathrm{MM}'(r)$, and define the rest as $\mathrm{MM}(r)$:

$$\left\|\int f(x;\phi)r(x|y;\omega)P_{bal}(x|y;\omega)\mathrm{d}x\right\|^2 - 2\left\langle\int f(x;\phi)r(x|y;\omega)P(x|y;\omega)\mathrm{d}x, \int f(x;\phi)P(x|y;\omega)\mathrm{d}x\right\rangle \tag{5}$$

where $\langle\cdot, \cdot\rangle$ denotes the inner product. In practice, as for the real-world imbalanced data distribution $P$, we denote $\mathbf{\Phi}_P$ to dynamically reflect the knowledge learned from the distribution $P$, that is $\mathbf{\Phi}_P = (f(x_1;\phi), \ldots, f(x_n;\phi))$. Remember that the output of feature extractor is $z$, *i.e.*, $z = f(x;\phi)$ is a $Z$-dimensional vector, then $\mathbf{\Phi}_P$ would be a $[Z, n]$-dimensional vector. Similarly, we denote $\mathbf{\Phi}_P^i$ for class $i$, which is a $[Z, n_i]$-dimensional vector. As for the balanced data distribution $P_{bal}$, we design a momentum mechanism to accumulatively estimate the expectation of features learned from balanced data distribution along with the training. Concretely, for each class, we maintain a prototype feature $F$ for the entire training progress, using each batch's feature expectation for momentum updates. Therefore, we define $F_{P_{bal}}$ as follows $F_{P_{bal}} = (F_1, \ldots, F_d)$.

Since the total number of classes is $d$, $F_{P_{bal}}$ would be a $[Z, d]$-dimensional vector. For each batch, we can obtain $\overline{z} = (\overline{z}_1, \ldots, \overline{z}_d)$, where $\overline{z}_i$ the mean of $z$ of all samples in class $i$. Then, the momentum updates works as follows

$$F_{P_{bal}} \leftarrow mF_{P_{bal}} + (1-m)\overline{z} \tag{6}$$

where $m \in [0, 1)$ is a momentum coefficient. Back to Eq. (5), replace the expectations over $P_{bal}$ and $P$ by $\mathbf{\Phi}_P$ and $F_{P_{bal}}$, respectively. Then, take the derivative of $\mathrm{MM}(r)$ with respect to $r$ and set it to zero. Detailed derivations are provided in Appendix C.1. For each class $i$, we can obtain the estimation of density ratio in imbalanced learning as follows

$$\widehat{r}_i = n_i\left(\mathbf{\Phi}_P^i{}^\top\mathbf{\Phi}_P^i\right)^{-1}\mathbf{\Phi}_P^i{}^\top F_i \tag{7}$$

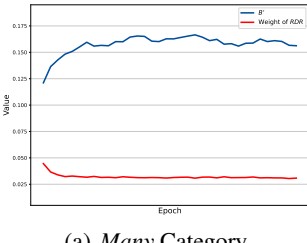 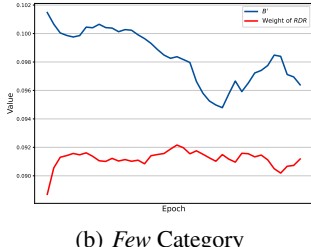

(a) *Many* Category          (b) *Few* Category

Figure 1: Dynamic trend of the RDR weights well inversely aligns with $B'$ throughout the training process in different categories. $B'_y$ denotes $\sqrt{\pi_y}\left[1 - \text{softmax}\left(B_y(m)\right)\right]$, where $B_y(m)$ denotes the minimal prediction on the ground-truth class $y$, i.e., $\min_{\boldsymbol{x} \in \mathcal{S}_y} m(\boldsymbol{x})_y$. Experiments were conducted on CIFAR-10-LT dataset with an imbalance factor of 10.

Substitute Eq. (7) into Eq. (1), we can obtain our object to optimize

$$
\overline{R} = \sum_{i=1}^{d} \frac{1}{\pi_i} \sum_{y_j = i} n_i \left(\boldsymbol{\Phi}_P^{i^\top} \boldsymbol{\Phi}_P^i\right)^{-1} \boldsymbol{\Phi}_P^{i^\top} F_i \cdot l(x_j, y_j; \omega)
$$
$$
\propto \sum_{i=1}^{d} \sum_{y_j = i} \left(\boldsymbol{\Phi}_P^{i^\top} \boldsymbol{\Phi}_P^i\right)^{-1} \boldsymbol{\Phi}_P^{i^\top} F_i \cdot l(x_j, y_j; \omega)
$$

(8)

In our implementation, we introduced a warm-up phase to pre-adapt the feature distribution in $F_{P_{bal}}$, thereby mitigating excessive oscillations during the initial stages of training. Additionally, we employed a temperature coefficient $\gamma$ to modulate the influence of weights, which is typically set to 1. When integrating with logit adjustment (LA) [Menon et al., 2020], we adhere to the same procedures outlined in [Wang et al., 2023] to ensure the fisher consistency. The framework and pseudo-code of our method are shown in Appendix C.2 and Appendix C.3.

### 3.4 Generalization Bound Analysis

Here, we use a formal generalization analysis to characterize the interesting point of our method.

**Theorem 1.** *Given a model $m \in \mathcal{M}$ and the loss function $l$, for any $\delta \in (0, 1)$, with probability at least $1 - \delta$ over the training set $\mathcal{S}$, according to [Wang et al., 2023], the following generalization bound holds for the risk on the balanced distribution*

$$
R_{bal}^l(m) \precsim \Phi\left(l, \delta\right) + \frac{\mathfrak{S}_{\mathcal{S}}(\mathcal{M})}{d\pi_1} \sum_{y=1}^{d} w_y \sqrt{\pi_y}\left[1 - \text{softmax}\left(B_y(m)\right)\right]
$$

(9)

*where $\Phi\left(l, \delta\right)$ is positively correlated with the empirical re-weighting risk of the training set. $\mathfrak{C}_{\mathcal{S}}(\mathcal{M})$ denotes the empirical complexity of the function set $\mathcal{M}$. $B_y(f)$ denotes the minimal prediction on the ground-truth class $y$ in the training set. $w_y$ refers to the weight of class $y$ of the re-weighting loss.*

Specifically, from the above generalization bound, we can find two inherent requirements for re-weighting methods. 1) *Why re-weighting is necessary*: $w_y$ helps to re-balance the imbalanced term $\sqrt{\pi_y}\left[1 - \text{softmax}\left(B_y(m)\right)\right]$ to get a sharper bound. 2) *Why dynamic re-weighting is necessary*: The term $B_y(m)$ changes dynamically with model training. Therefore, we need a $w_y$ that can adapt dynamically to the changes of $B_y(m)$. 3) *Why RDR works*: From Fig. 1, we can observe that the dynamic trend of the RDR weight aligns well with $\sqrt{\pi_y}\left[1 - \text{softmax}\left(B_y(m)\right)\right]$, denoted as $B'_y$. This shows that our RDR can adapt to the dynamics in $B'_y$, maintaining a sharp bound during training.

### 3.5 Implementation and Complexity Analysis

At the end of each epoch, a global variable is maintained and updated using momentum, as described by Eq. (6). Within each minibatch, the weight of each sample is computed dynamically. Typical

optimization procedures for deep neural networks entail both forward and backward passes per mini-batch, characterized by a computational complexity of $\mathcal{O}(B\Lambda)$, where $B$ represents the batch size and $\Lambda$ denotes the overall parameter size. Within the RDR framework, suppose the feature dimension used as input to the classifier is $K$, and the sample weights are computed according to Eq. (8). This computation for all $d$ classes aggregates to a complexity of $\mathcal{O}(\sum_{i=1}^{d}(n_i \times K^2 + K^3))$ where $n_i$ is the sample count of class $i$ in a minibatch. Given that $K$ generally exceeds $n_i$, the complexity predominantly stems from the matrix inversion, approximating to $\mathcal{O}(dK^3)$. The complexity for momentum updates is $\mathcal{O}(BK)$. Notice that $K$ and $B$ are considerably minor relative to the scale of the model parameters, rendering the time overhead of this method manageable.

On the storage front, the memory cost of the RDR primarily arises from the matrix inversion step in Eq. (8), resulting in a space complexity of $\mathcal{O}(K^2)$. Given the scales of $K$ is much lower than $\Lambda$, the extra memory usage is negligible when compared with the memory utilization of the model parameters. To this end, RDR imposes a relatively small computational or space cost, enabling its integration with existing approaches at a reduced cost. An empirical evaluation of the computational expense is presented in Fig. 2. For more discussions about limitations of RDR , please refer to Appendix E.

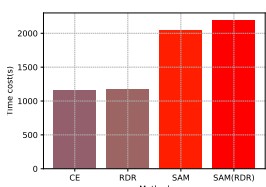
(a) Time Cost on CIFAR-10-LT

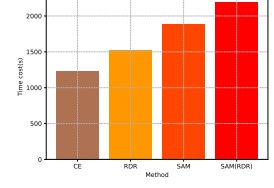
(b) Time Cost on CIFAR-100-LT

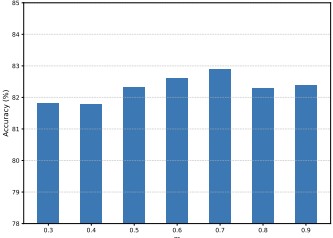

Figure 2: Visualization of the time cost for training 200 epochs using four methods: CE, RDR, SAM and RDR(SAM) on CIFAR-10-LT and CIFAR-100-LT datasets.

Figure 3: The impact of momentum coefficient $m$ in RDR under the measure of top-1 accuracy.

## 4 Ex:priments

### 4.1 Experimental Setup

**Datasets.** We conduct experiments on four major long-tailed datasets, CIFAR-10-LT, CIFAR-100-LT, ImageNet-LT [Liu et al., 2019] and Places-LT [Liu et al., 2019]. CIFAR-10-LT and CIFAR-100-LT are two datasets sampled from the original CIFAR [Krizhevsky et al., 2009] dataset with a total of 10 and 100 classes, respectively. We conduct experiments with different imbalance factors $IF = \frac{n_{max}}{n_{min}}$, where $n_{max}$ and $n_{min}$ denotes the number of the most and least frequent classes [Kang et al., 2020, Hong et al., 2023, 2024b]. Following the mainstream protocol [Wang et al., 2023], we set the imbalance factor as 100 and 10 for evaluation. ImageNet-LT has 115.8K training images covering 1000 classes, with imbalance factor being 256. The number of samples per class ranges from 1280 to 5 images. Places-LT contains 62.5K training images covering 365 categories, with imbalance factor being 996. The number of samples per class ranges from 4980 to 5 images.

**Evaluation Protocol.** In the task of long-tailed classification, all classes are treated equally during testing. Following [Rangwani et al., 2022, Zhou et al., 2023c], we also report accuracy on three splits of classes according to the number of training data. Since the number of samples per class increases by its class index, for CIFAR-10-LT dataset, class[0, 3), class[3, 7) and class[7, 10) are reported as *Many*, *Medium* and *Few* classes, respectively. Similarly, CIFAR-100-LT is splited as class[0, 35), class[35, 69) and class[69, 100). ImageNet-LT is splited as class[0, 390), class[390, 835) and class[835, 1000), while Places-LT is splited as class[0, 131), class[131, 288) and class[288, 365).

**Baselines.** Our method is combined with existing long-tailed classification methods to demonstrate the efficacy, including the baseline trained by cross-entropy loss (CE), focal loss (Focal) [Lin et al., 2017], class-balanced loss (CB) [Cui et al., 2019] and logit adjustment (LA) [Menon et al., 2020]. Recently, Sharpness-Aware minimization (SAM) [Foret et al., 2021] has been proved to be a powerful method in imbalanced learning, therefore we also adopt baseline including SAM [Foret et al., 2021],

Table 1: Top-1 accuracy (%) (↑) results for overall classes on CIFAR-10-LT, CIFAR-100-LT, ImageNet-LT and Places-LT, CIFAR-10-LT and CIFAR-100-LT are employed with imbalance factors of 10 and 100, respectively.

| Dataset | CIFAR-10-LT | | CIFAR-100-LT | | ImageNet-LT | Places-LT |
| | IF=10 | IF=100 | IF=10 | IF=100 | | |
|---|---|---|---|---|---|---|
| CE | $88.9_{\pm0.4}$ | $75.6_{\pm0.8}$ | $59.3_{\pm0.7}$ | $42.7_{\pm0.3}$ | $43.2_{\pm0.1}$ | $29.3_{\pm0.2}$ |
| Focal | $89.0_{\pm0.3}$ | $76.0_{\pm0.1}$ | $59.7_{\pm0.5}$ | $43.0_{\pm0.6}$ | $43.8_{\pm0.8}$ | $29.5_{\pm0.2}$ |
| CB | $89.0_{\pm0.4}$ | $76.7_{\pm0.8}$ | $60.4_{\pm0.6}$ | $43.5_{\pm1.2}$ | $43.8_{\pm0.1}$ | $32.5_{\pm0.3}$ |
| LA | $89.2_{\pm0.3}$ | $82.2_{\pm0.7}$ | $62.3_{\pm0.5}$ | $48.2_{\pm0.4}$ | $47.9_{\pm0.4}$ | $37.5_{\pm0.2}$ |
| RDR+CE | $89.9_{\pm0.1}$ | $81.9_{\pm0.1}$ | $62.3_{\pm0.4}$ | $48.5_{\pm0.4}$ | $45.2_{\pm0.1}$ | $39.4_{\pm0.2}$ |
| RDR+LA | $\mathbf{90.2}_{\pm0.4}$ | $\mathbf{83.4}_{\pm0.3}$ | $\mathbf{62.9}_{\pm0.2}$ | $\mathbf{49.4}_{\pm0.3}$ | $\mathbf{48.1}_{\pm0.3}$ | $\mathbf{39.5}_{\pm0.1}$ |

ImbSAM [Zhou et al., 2023a] and CCSAM [Zhou et al., 2023c], the latter two are also SAM-based methods. The strategies above have been demonstrated superior performance in imbalanced learning.

**Implementation details.** Our code is implemented with Pytorch 1.12.1. Experiments based on CIFAR-10-LT and CIFAR-100-LT are carried out on NVIDIA GeForce RTX 3090 GPUs, while experiments based on ImageNet-LT and Places-LT are carried out on NVIDIA A100 GPUs. For a fair comparison, we use ResNet32 on CIFAR-10-LT and CIFAR-100-LT, ResNet50 on ImageNet-LT and pre-trained ResNet-152 on Places-LT. We train each model with batch size of 128 (for CIFAR-10-LT and CIFAR-100-LT) / 256 (for Places-LT and ImageNet-LT), SGD optimizer with momentum of 0.9, weight decay of 0.0002. The initial learning rate is set to 0.1, with cosine learning-rate scheduling along training. The results of ImbSAM and CCSAM are obtained by implementing the official codes.

## 4.2 Comparison Results

Comparative analyses have been performed to evaluate the effectiveness of the proposed RDR. The results are presented in Table 1, Table 2 and Table 4. The metric employed to measure performance is the top-1 accuracy on the test sets.

**Results on CIFAR-10-LT and CIFAR-100-LT.** We first evaluate RDR on CIFAR-10-LT and CIFAR-100-LT. We report the final accuracy of different methods with imbalance factor ratio {10, 100} in Table 1 and Table 2. We can observe that RDR significantly outperforms all baselines under different imbalance factor ratios across the two datasets. Our observations highlight that RDR consistently outperforms baselines across various class distributions—*Many*, *Medium*, and *Few*—particularly under severe imbalance (*IF*=100).

In CIFAR-10-LT, combined with CE and LA, our method shows substantial improvement in the categories with fewer samples, increasing the accuracy by 19.8% and 9.5% respectively in the Few category under *IF*=100. This improvement is notable as it effectively addresses the challenge of learning from scarce data. With the inclusion of SAM, the performance of RDR is further enhanced. Under a less severe imbalance (*IF*=10), where the results show less performance drop-off between categories, RDR combined methods still maintain high performance across all categories, suggesting scalability and reliability of our approach in different imbalance contexts.

In CIFAR-100-LT, where the data distributions are more diverse and challenging, RDR also enhances the overall performance, particularly for the *Medium* and *Few* categories. Under the imbalance factor of 10 and 100, RDR increases the accuracy in *Few* classes by 11.8% and 20.0% respectively, compared to the original CE loss. Furthermore, it is notable that techniques like ImbSAM and CCSAM, which especially focus on the *Few* categories, may heavily sacrifice the performance on *Many* classes. The results in both datasets show that RDR generally outperforms in the *Many* classes compared to the other two variants of SAM, indicating that RDR can efficiently address the overfitting issues for *Few* classes. For more experimental details, please refer to Appendix D.2 and Appendix D.1.

**Flat minima of loss landscape.** Key metrics associated with Eigen Spectral Density, such as the maximum and minimum eigenvalues ($\lambda_{max}$ and $\lambda_{min}$) and the trace of the Hessian matrix ($Tr(H)$), effectively reflect the smoothness of the loss landscape. Lower values of $\lambda_{max}$ and $Tr(H)$ indicate a smoother loss landscape. Rangwani et al. [2022] have demonstrated that smoother loss landscapes correlate with stronger model generalization, which is particularly crucial when dealing

Table 2: Top-1 accuracy (%) (↑) results for *Many*, *Medium*, *Few* and overall classes on CIFAR-10-LT, categorized by imbalance factors (*IF*) of 100 and 10. The experiments are employed with the integration of Sharpness-Aware-Minimization-based methods.

| Dataset | Loss | Method | IF=100 | | | | IF=10 | | | |
|---|---|---|---|---|---|---|---|---|---|---|
| | | | Many | Med. | Few | All | Many | Med. | Few | All |
| CIFAR-10-LT | CE | SAM | 94.8 | 74.5 | 60.6 | 76.4 | 95.8 | 85.9 | 86.5 | 89.3 |
| | | ImbSAM | 94.6 | 73.9 | 67.7 | 78.3 | 94.2 | 84.1 | 90.1 | 89.0 |
| | | CCSAM | 85.6 | 79.2 | 80.3 | 81.4 | 90.6 | 85.3 | 90.8 | 88.5 |
| | | RDR+SAM | 91.8 | 78.6 | 81.9 | **83.6** | 94.2 | 86.3 | 92.5 | **90.5** |
| | LA | SAM | 90.8 | 78.1 | 82.9 | 83.3 | 92.6 | 86.5 | 91.8 | 89.9 |
| | | ImbSAM | 85.2 | 75.1 | 89.6 | 82.5 | 90.8 | 83.6 | 94.8 | 89.1 |
| | | CCSAM | 85.8 | 77.8 | 80.5 | 81.0 | 90.7 | 82.4 | 90.8 | 87.4 |
| | | RDR+SAM | 88.5 | 80.0 | 85.0 | **84.1** | 92.5 | 86.5 | 93.6 | **90.4** |
| CIFAR-100-LT | CE | SAM | 72.7 | 40.3 | 7.5 | 41.5 | 75.3 | 60.2 | 45.1 | 60.8 |
| | | ImbSAM | 71.5 | 40.6 | 17.7 | 44.3 | 72.5 | 57.9 | 53.5 | 61.7 |
| | | CCSAM | 61.5 | 50.8 | 29.7 | 48.0 | 62.8 | 59.3 | 54.5 | 59.0 |
| | | RDR+SAM | 63.4 | 51.9 | 30.5 | **49.3** | 68.0 | 61.6 | 58.3 | **62.8** |
| | LA | SAM | 64.3 | 50.5 | 30.8 | 49.2 | 66.1 | 60.9 | 59.7 | 62.3 |
| | | ImbSAM | 58.8 | 45.4 | 40.1 | 48.4 | 62.9 | 56.6 | 64.2 | 61.2 |
| | | CCSAM | 57.7 | 48.9 | 29.0 | 45.8 | 61.0 | 57.8 | 51.9 | 57.1 |
| | | RDR+SAM | 63.9 | 52.4 | 30.5 | **49.6** | 67.6 | 61.7 | 60.1 | **63.2** |

(a) CE      (b) RDR      (c) SAM      (d) RDR(SAM)

Figure 4: Eigen spectral density for the class with the fewest samples across different methods. Experiments conduct on the CIFAR-10-LT, under an imbalance factor of 100. Maximum eigenvalue $\lambda_{max}$ (↓) minimum eigenvalue $\lambda_{min}$ (↑) in the top right corner of each panel. A lower $\lambda_{max}$ indicates a smoother loss landscape, while a higher $\lambda_{min}$ suggests conditions more favorable for escaping from saddle points, thereby enhancing the model's generalization capabilities.

with imbalanced data. $\lambda_{min}$ also serves as a significant indicator of the loss landscape characteristics. A preponderance of negative eigenvalues from the Hessian spectrum, resulting in smaller $\lambda_{min}$ values, empirically suggests convergence to saddle points. Saddle points typically represent regions in the loss landscape characterized by a plateau with some negative curvature. In non-convex settings, it has been shown that an exponential number of saddle points exist, and convergence to these points is indicative of poor generalization.

Fig. 4 illustrates the Eigen Spectral Density under different loss function training regimes. It is evident that combining our method with the CE technique significantly improves the loss landscape. On one hand, $\lambda_{max}$ is substantially reduced, indicating a flatter loss landscape. On the other hand, there is an increase in $\lambda_{min}$, suggesting our method's effectiveness in escaping from saddle points.

Table 3 delves deeper into the changes in the loss landscape for all *Few* classes. It reveals that combining our method with CE and SAM results in average reductions in $\lambda_{max}$ by 58.5% and 66.1%, respectively, and increases in $\lambda_{min}$ by 49.2% and 7.6%, respectively. Furthermore, our method significantly reduces $Tr(H)$ for minority classes (class7, class8 and class9). The average value of $Tr(H)$ decreases by 55.7% and 66.0% when combined with CE and SAM, respectively. These findings underscore the efficacy of our method in improving the loss landscape and enhancing the generalization capability of the model [Dauphin et al., 2014].

**Results on ImageNet-LT and Places-LT.** Our experiments conducted on ImageNet-LT and Places-LT, two large-scale datasets characterized by irregular and complex data distributions, demonstrate notable accuracy improvements through the application of RDR, as shown in Table 1 and Table 4.

Table 3: Loss landscape metrics across different methods on CIFAR-10-LT, with imbalance factor 100. Average minimum eigenvalues $\overline{\lambda_{\min}}$ ($\uparrow$), average maximum eigenvalues $\overline{\lambda_{\max}}$ ($\downarrow$), and the trace $Tr$ ($\downarrow$) of the Hessian matrix for classes with few samples. $Tr_6$, $Tr_7$, $Tr_8$, and $Tr_9$ represent the traces of the Hessian matrix for class 6, 7, 8 and 9, respectively, with descending sample quantities. $\overline{Tr_{Few}}$ denotes average trace of Hessian matrix over *Few* classes. Lower $\lambda_{\max}$ and $Tr$ values indicate a flatter loss landscape, while a higher $\lambda_{\min}$ suggests a landscape more conducive to escaping from saddle points, thereby potentially enhancing model generalization.

| Method | $\overline{\lambda_{\min}}$ | $\overline{\lambda_{\max}}$ | $Tr_6$ | $Tr_7$ | $Tr_8$ | $Tr_9$ | $\overline{Tr_{Few}}$ |
|--------|------|------|------|------|------|------|------|
| SGD | -76.24 | 110.78 | 335.64 | 301.67 | 234.97 | 323.87 | 286.84 |
| RDR | -38.73 | 45.95 | 290.06 | 293.04 | 46.15 | **41.71** | 126.96 |
| SAM | -14.42 | 74.82 | 300.13 | 314.48 | 123.10 | 291.00 | 242.86 |
| RDR+SAM | **-13.33** | **25.36** | **179.50** | **147.11** | **36.28** | 64.32 | **82.57** |

Specifically, when combined with CE and LA on ImageNet-LT, our method achieves accuracy enhancements of 18.7% and 2% in *Few* classes, respectively. While the SAM technique combined with CE and LA offers limited accuracy improvements, its integration with our approach still results in an overall accuracy increase of approximately 2.3% compared to other methods. Notably, the LA method tends to suppress accuracy in *Many* classes more than the CE method; however, this side effect is effectively mitigated when LA is combined with RDR.

Table 4: Top-1 accuracy (%) ($\uparrow$) results for *Many*, *Medium*, *Few* and overall classes on ImageNet-LT and Places-LT. The experiments are employed with the integration of Sharpness-Aware-Minimization-based methods.

| Loss | Method | ImageNet-LT | | | | Places-LT | | | |
|------|--------|------|------|------|------|------|------|------|------|
| | | Many | Med. | Few | All | Many | Med. | Few | All |
| CE | SAM | 64.6 | 35.8 | 10.1 | 42.8 | 45.2 | 27.6 | 12.1 | 30.6 |
| | ImbSAM | 62.5 | 37.3 | 13.9 | 43.3 | 43.1 | 26.2 | 16.1 | 30.1 |
| | CCSAM | 54.1 | 44.1 | 30.8 | 45.8 | 40.4 | 39.7 | 31.7 | 38.2 |
| | RDR+SAM | 59.1 | 46.5 | 26.2 | **48.1** | 41.5 | 41.3 | 39.0 | **40.9** |
| LA | SAM | 39.6 | 45.9 | 39.1 | 42.3 | 42.5 | 41.9 | 35.5 | 40.8 |
| | ImbSAM | 56.1 | 45.1 | 37.9 | 48.2 | 38.6 | 35.8 | 40.4 | 37.8 |
| | CCSAM | 52.2 | 43.9 | 32.8 | 45.3 | 32.1 | 41.4 | 40.9 | 37.9 |
| | RDR+SAM | 57.5 | 49.7 | 35.9 | **50.5** | 41.5 | 42.9 | 37.8 | **41.3** |

In the Places-LT dataset, the performance of RDR is even more pronounced. Combinations of our method with CE and LA result in accuracy gains of 10.1% and 2% in overall classes, respectively. Additionally, integrating SAM with our method also yields incremental improvements of 10.3% and 0.5% under CE and LA conditions, respectively. Our approach not only enhances accuracy in *Few* classes but also surpasses other methods in *Medium* classes, indicating its comprehensive efficacy across different categories. This broad applicability is particularly crucial for addressing the challenges of imbalanced learning.

**Results on data with label noise.** We further investigate the performance of our approach on datasets with label noise. Specifically, we evaluate two datasets: CIFAR-10-LT-NL and CIFAR-100-LT-NL, both of which exhibit class imbalance and label noise. Experiments are conducted with a noise ratio of 5%, and the results are presented in Fig. 5. As shown, our method demonstrates consistent and significant improvements on more datasets with label noise.

## 4.3 Ablation Study

In our study, we perform an ablation experiment to validate the efficacy of the multiple components that comprise our method. The outcomes from experiments across four datasets are delineated in Table 5. It is crucial to note that our weighting definition for each category $i$ follows the formula $w = r/n_i$. When $r = 1$, our method simplifies to the traditional inverse frequency weighting $w = 1/n_i$. We explored the differences between our approach with and without the integration of SAM compared to this conventional weighting method.

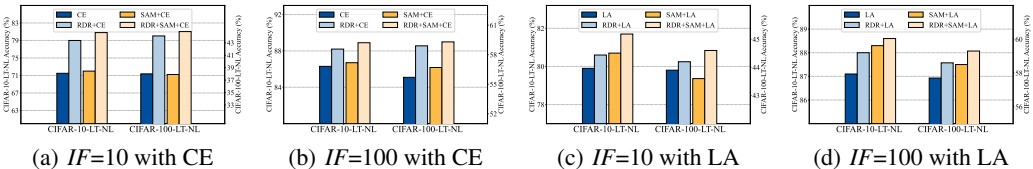

| (a) *IF*=10 with CE | (b) *IF*=100 with CE | (c) *IF*=10 with LA | (d) *IF*=100 with LA |

Figure 5: Top-1 accuracy (%) (↑) results for overall classes on CIFAR-10-LT-NL and CIFAR-100-LT-NL with 5% noise ratio, categorized by imbalance factors (*IF*) of 100 and 10.

Table 5: Top-1 accuracy (%) (↑) results from ablation studies across diverse datasets. Experiments conduct on CIFAR-10-LT, CIFAR-100-LT, ImageNet-LT, and Places-LT, comparing different method combinations. $1/n$ denote classic inverse frequency weighting method, assigning weights of $1/n_i$ for class $i$.

| Loss | Method | | | CIFAR-10-LT | | CIFAR-100-LT | | ImageNet-LT | Places-LT |
|---|---|---|---|---|---|---|---|---|---|
| | $1/n$ | RDR | SAM | IF=100 | IF=10 | IF=100 | IF=10 | | |
| CE | | | | 75.61 | 88.86 | 42.66 | 60.17 | 43.18 | 29.27 |
| | ✓ | | | 77.78 | 89.50 | 44.21 | 61.89 | 44.62 | 34.27 |
| | | ✓ | | 81.87 | 89.94 | 48.54 | 62.28 | 45.19 | 35.45 |
| | | | ✓ | 76.44 | 89.33 | 41.46 | 60.82 | 42.78 | 30.64 |
| | ✓ | | ✓ | 80.98 | 90.22 | 44.82 | 61.43 | 47.47 | 38.73 |
| | | ✓ | ✓ | **82.88** | **90.52** | **49.30** | **62.80** | **48.06** | **48.06** |
| LA | | | | 82.15 | 89.17 | 48.34 | 62.27 | 47.86 | 37.51 |
| | ✓ | | | 82.27 | 89.27 | 48.55 | 62.26 | 45.86 | 37.48 |
| | | ✓ | | 83.44 | 90.21 | 49.35 | 62.88 | 48.12 | 37.79 |
| | | | ✓ | 83.37 | 89.89 | 49.21 | 62.34 | 42.34 | 40.77 |
| | ✓ | | ✓ | 82.71 | 90.25 | 47.36 | 61.88 | 43.40 | 39.48 |
| | | ✓ | ✓ | **83.56** | **90.41** | **49.63** | **63.06** | **50.45** | **41.33** |

The results depicted in Table 5 reveal that our dynamic weighting approach consistently outperforms the classic method under various scenarios. Without SAM, when combined with CE and LA, our method achieves accuracy improvements ranging from 0.4% to 4.3% and 0.3% to 2.3%, respectively. When integrated with SAM, the improvement in accuracy is particularly notable on large datasets. Specifically, the accuracy enhancements on ImageNet-LT and Places-LT reach 7.1% and 1.9% respectively when combined with LA. These results underscore the tangible benefits of our dynamic weighting strategy in enhancing model performance. The impact of momentum coefficient $m$ in RDR is shown in Fig. 3. For more experimental details, please refer to Appendix D.3.

# 5 Conclusion

In this work, we have introduced RDR, a novel approach for mitigating model degradation in imbalanced learning scenarios by dynamically adjusting class weights using density ratio estimation. Our method dynamically adjusts class weights during training based on density ratio estimation, enhancing both model robustness and adaptability. Extensive experiments on diverse large-scale datasets demonstrate the effectiveness of RDR, particularly in severely imbalanced settings. Future work will focus on refining the dynamic adjustment mechanisms and exploring broader applicability across various domains and dataset complexities.

# Acknowledgement

Jiaan Luo, Feng Hong, Jiangchao Yao, Ya Zhang and Yanfeng Wang are supported by the National Key R&D Program of China (No. 2022ZD0160702), STCSM (No. 22511106101, No. 22DZ2229005), 111 plan (No. BP0719010) and National Natural Science Foundation of China (No. 62306178). Bo Han is supported by NSFC General Program No. 62376235, Guangdong Basic and Applied Basic Research Foundation No. 2022A1515011652 and No. 2024A1515012399, HKBU Faculty Niche Research Areas No. RC-FNRA-IG/22-23/SCI/04 and HKBU CSD Departmental Incentive Scheme.

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

# A  Notations

In Table 6, we summarize the notations used in this paper.

Table 6: Description of Notations

| Category | Notation | Description |
|---|---|---|
| Data and Sets | $\mathcal{S}$ | Training dataset |
| | $\pi_i$ | Proportion of class $i$ |
| | $n_i$ | Sample number of class $i$ |
| | $d$ | Number of classes |
| | $B$ | Training batch size |
| Model and Functions | $f(\cdot; \phi)$ | Feature extractor with parameters $\phi$ |
| | $h(\cdot; \theta)$ | Classifier with parameters $\theta$ |
| | $\omega = \bigcup\{\phi, \theta\}$ | Model parameters consist of $\phi$ and $\theta$ |
| | $l(x, y; \omega)$ | Loss function with input $x$, output $y$ and parameters $\omega$ |
| | $R$ | Empirical risk of the loss function |
| | $P$ | Distribution on training set |
| | $P_{bal}$ | Distribution on balanced data set |
| | $r$ | Density ratio |
| | $\mathbf{\Phi}_P$ | Matrix of knowledge learned from the distribution $P$ |
| | $\mathbf{\Phi}_P^i$ | $\mathbf{\Phi}_P$ for class $i$ |
| | $F_{P_{bal}}$ | Matrix of feature expectation for momentum update |
| | $Z$ | Dimension of features extracted from feature extractor |
| | $B_y(\cdot)$ | Minimal prediction on the ground-truth class $y$ |
| | $m \in M$ | Model $m$ in function set $M$ |
| Others | $IF$ | Imbalance factor |
| | $\overline{\lambda_{max}}, \overline{\lambda_{min}}$ | Average maximum and minimum eigenvalues |
| | $Tr_i$ | Trace of Hessian matrix for class $i$ |
| | $\overline{Tr}_{Few}$ | Average trace of Hessian matrix over *Few* categories |

# B  More Discussions of Related Work

Beyond imbalanced classification that has discretized label space, an noteworthy area, imbalanced regression that has a continuous label space is also very common in real applications [Yang et al., 2021, Gong et al., 2022]. In this direction, the empirical label distribution often does not accurately reflect the true label density in regression tasks, which limits the effectiveness of traditional re-weighting techniques [Yang et al., 2021, Wang and Wang, 2023]. Label Distribution Smoothing (LDS) [Yang et al., 2021] and Variational Imbalanced Regression (VIR) [Wang and Wang, 2023] propose using kernel smoothing and other techniques to estimate an accurate label density distribution. Ranking Similarity (Ranksim) [Gong et al., 2022] leverages local and global dependencies by encouraging the correspondence between the similarity order of labels and features. Balanced Mean Squared Error (Balanced MSE) [Ren et al., 2022] extends the concept of Balanced Softmax [Ren et al., 2020a] to regression tasks to achieve a balanced predictive distribution. Contrastive Regularizer (ConR) [Keramati et al., 2024] improves contrastive learning techniques to translate label similarities into the feature space.

# C  Algorithm Details

## C.1  Detailed Derivations of Eq. (7)

Back to Eq. (5), replace the expectations over $P_{bal}$ and $P$ by $\mathbf{\Phi}_P$ and $F_{P_{bal}}$, respectively. For each class $i$, We can obtain $\widehat{r_i} = \widehat{\mathrm{MM}}(r)$, where

$$\widehat{\mathrm{MM}}(r) = \frac{1}{n_i^2} r_i^\top \mathbf{\Phi}_P^{i\ \top} \mathbf{\Phi}_P^i r_i - \frac{2}{n_i} r_i^\top \mathbf{\Phi}_P^{i\ \top} F_i \tag{10}$$

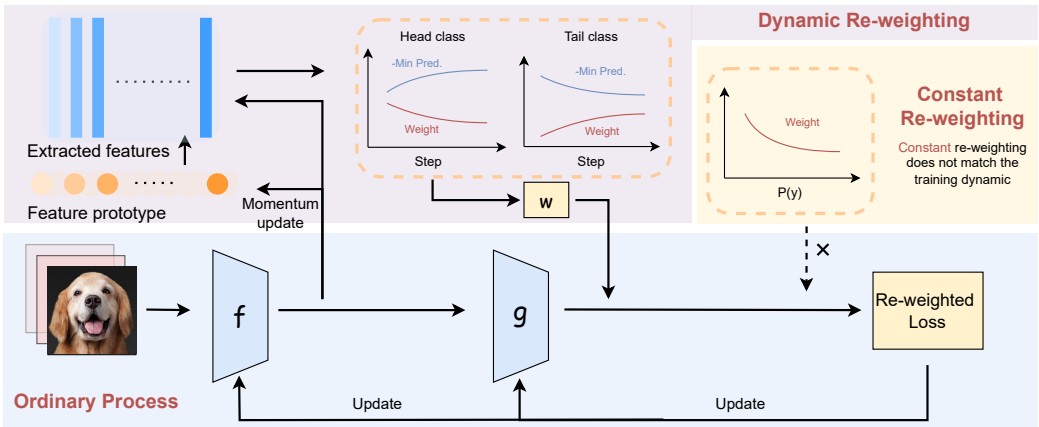

Figure 6: Framework of RDR

Then, taking the derivative of $\widehat{\mathrm{MM}}(r)$ with respect to $r$ and setting it to zero, we can obtain the estimation of density ratio in imbalanced learning as follows

$$\frac{2}{n_i^2}{\mathbf{\Phi}_P^i}^\top \mathbf{\Phi}_P^i r_i - \frac{2}{n_i}{\mathbf{\Phi}_P^i}^\top F_i = 0 \tag{11}$$

Solving equation above with respect to $r_i$, we can obtain the solution as

$$\widehat{r_i} = n_i \left({\mathbf{\Phi}_P^i}^\top \mathbf{\Phi}_P^i\right)^{-1} {\mathbf{\Phi}_P^i}^\top F_i \tag{12}$$

## C.2 Framework of RDR

We provide the framework of RDR, which is shown in Fig. 6.

## C.3 Pseudo-code of RDR

We provide the pseudo-code of RDR to demonstrate the process of implementing our method in detail, as shown in Algorithm 1. In addition, we also provide pseudo-code that combines our method with the SAM method, as shown in Algorithm 2.

# D Supplement for Experiments

## D.1 Experiment with More Imbalanced Data

We conduct experiments on the more imbalanced CIFAR-10-LT and CIFAR-100-LT datasets, specifically with imbalance factors of 200 and 500. As shown in Table 7, our method consistently achieves significant improvements.

## D.2 Dynamically Re-weighting Process

Our approach dynamically adjusts the weights assigned to each category throughout the training process. To gain more insights into RDR, we sampled the weights of each category during training. Fig. 7 presents the results of four samplings during the training processes at imbalance factors of 10 and 100, respectively.

The analysis of these results reveals a consistent trend in weight changes across different imbalance factors. For the *Many* classes, the weights of the categories consistently decrease during training. Specifically, under the *IF* of 10, the weights of class0, class1, and class2 (the three classes with the highest sample counts, in descending order) decrease by 5.6%, 6.7%, and 8.0%, respectively. Under the *IF* of 100, these decreases are more pronounced, with reductions of 10.3%, 15.4%, and 14.7%, respectively. For the *Medium* classes, the weight changes are less marked, with an average decrease

---

**Algorithm 1** Training Paradigm of RDR.

---

1: **Input:** Training dataset $\mathcal{S} = \cup_{i=1}^n \{(x_i, y_i)\}$, model $\mathcal{M}_\omega$ with feature extractor $f_\phi$ and classifier $h_\theta$, loss function $l$, momentem coefficient $m$, learning rate $\alpha$, weight decay coefficient $\lambda$, batch size $b$, temperature coefficient $\gamma$
2: **Output:** Trained parameters $\phi^*$, $\theta^*$
3: Initialize the model parameters $\phi$ and $\theta$ ramdomly, $F_{P_{bal}} = (F_1, \ldots, F_d) \leftarrow 0$
4: **for** $t = 1$ to $T$ **do**
5: $\quad \mathcal{B} \leftarrow$ SampleMiniBatch$(\mathcal{S}, b)$
6: $\quad z \leftarrow f(x, \phi_t)$
7: $\quad output \leftarrow h(z, \theta_t)$
8: $\quad$ **if** $t < T_0$ **then**
9: $\quad\quad$ // `warm up`
10: $\quad\quad w \leftarrow 1$
11: $\quad$ **else**
12: $\quad\quad$ **for** class $i$ to $d$ **do**
13: $\quad\quad\quad \Phi_P^i \leftarrow (z_j)$ where $y_j = i$
14: $\quad\quad\quad$ compute $w_i$ via Eq. (8) and $\gamma$
15: $\quad\quad$ **end for**
16: $\quad\quad w \leftarrow$ normalize$(w^\gamma)$
17: $\quad$ **end if**
18: $\quad \mathcal{L}(\omega_t, \mathcal{B}) \leftarrow \frac{1}{b} \sum_{\mathcal{B}} w \cdot l(output, y)$
19: $\quad \omega_t = \omega_t - \alpha_t [\nabla \mathcal{L}(\omega_t, \mathcal{B}) + \lambda \omega_t]$
20: $\quad F_{P_{bal}} \leftarrow m F_{P_{bal}} + (1 - m)\bar{z}$
21: $\quad$ Optional: anneal the learning rate $\alpha_t$
22: **end for**

---

---

**Algorithm 2** Training Paradigm of RDR combined with SAM.

---

1: **Input:** Training dataset $\mathcal{S} = \cup_{i=1}^n \{(x_i, y_i)\}$, model $\mathcal{M}_\omega$ with feature extractor $f_\phi$ and classifier $h_\theta$, loss function $l$, momentem coefficient $m$, learning rate $\alpha$, weight decay coefficient $\lambda$, batch size $b$, neighborhood size $\rho$, temperature coefficient $\gamma$
2: **Output:** Trained parameters $\phi^*$, $\theta^*$
3: Initialize the model parameters $\phi$ and $\theta$ ramdomly, $F_{P_{bal}} = (F_1, \ldots, F_d) \leftarrow 0$
4: **for** $t = 1$ to $T$ **do**
5: $\quad \mathcal{B} \leftarrow$ SampleMiniBatch$(\mathcal{S}, b)$
6: $\quad z \leftarrow f(x, \phi_t)$
7: $\quad output \leftarrow h(z, \theta_t)$
8: $\quad$ **if** $t < T_0$ **then**
9: $\quad\quad$ // `warm up`
10: $\quad\quad w \leftarrow 1$
11: $\quad$ **else**
12: $\quad\quad$ **for** class $i$ to $d$ **do**
13: $\quad\quad\quad \Phi_P^i \leftarrow (z_j)$ where $y_j = i$
14: $\quad\quad\quad$ compute $w_i$ via Equation 8 and $\gamma$
15: $\quad\quad$ **end for**
16: $\quad\quad w \leftarrow$ normalize$(w^\gamma)$
17: $\quad$ **end if**
18: $\quad \mathcal{L}_1(\omega_t, \mathcal{B}) \leftarrow \frac{1}{b} \sum_{\mathcal{B}} w \cdot l(output, y)$
19: $\quad \epsilon_t \leftarrow \rho \frac{\nabla \mathcal{L}_1(\omega_t, \mathcal{B})}{|\nabla \mathcal{L}_1(\omega_t, \mathcal{B})|}$
20: $\quad \mathcal{L}_2(\omega_t + \epsilon_t, \mathcal{B}) \leftarrow \frac{1}{b} \sum_{\mathcal{B}} w \cdot l(f_{\omega_t + \epsilon_t}(\cdot), y)$
21: $\quad \omega_t = \omega_t - \alpha_t [\nabla \mathcal{L}_2(\omega_t + \epsilon_t, \mathcal{B}) + \lambda \omega_t]$
22: $\quad F_{P_{bal}} \leftarrow m F_{P_{bal}} + (1 - m)\bar{z}$
23: $\quad$ Optional: anneal the learning rate $\alpha_t$
24: **end for**

---

Table 7: Top-1 accuracy (%) (↑) results under more imbalanced conditions.

| Method | IF=500 | | IF=200 | |
|---|---|---|---|---|
| | CIFAR-10-LT | CIFAR-100-LT | CIFAR-10-LT | CIFAR-100-LT |
| CE | 60.1 | 34.1 | 68.6 | 37.9 |
| RDR+CE | 69.7 | 39.0 | 76.7 | 43.1 |
| SAM+CE | 60.4 | 34.5 | 69.9 | 39.0 |
| RDR+SAM+CE | **71.1** | **40.0** | **80.0** | **44.8** |
| LA | 73.9 | 38.8 | 76.7 | 44.0 |
| RDR+LA | 75.8 | 40.1 | 80.8 | 45.0 |
| SAM+LA | 74.9 | 39.6 | 80.2 | 44.2 |
| RDR+SAM+LA | **76.4** | **40.6** | **81.6** | **45.1** |

Figure 7: Visualization of weight changes across different categories throughout the training process. Experiments conducted on CIFAR-10-LT. (a), (b), and (c) illustrate the weight changes for *Many*, *Medium*, and *Few* classes, respectively, under an imbalance factor of 10. (d), (e), and (f) correspondingly show the changes for each class type under an imbalance factor of 100.

of 3.7% at *IF*=10 and 0.3% at *IF*=100. For the *Few* classes, there exits a notable increase in weights during training; at *IF*=10, the weights of class7, class8, and class9 increase by 2.3%, 6.3%, and 9.4%, respectively, while at *IF*=100, they increase by 6.2%, 6.5%, and 3.2%.

These results suggest that our method increasingly focuses on minority classes as training progresses. Initially, our method effectively learns common features across all categories, while later in training, increasing the weights helps to target learning towards minority samples, thereby enhancing the model's generalizability. Wang et al. [2023] also corroborate these findings.

## D.3 Ablation Study

We provide more detailed experimental results for each category in ablation study, as illustrated in Fig. 8 and Fig. 9. From these figures, we can find that across various datasets and different imbalance factors, our method significantly enhances the generalizability of both *Few* classes and *Medium* classes. Moreover, our method maintains superior performance when combined with the SAM method.

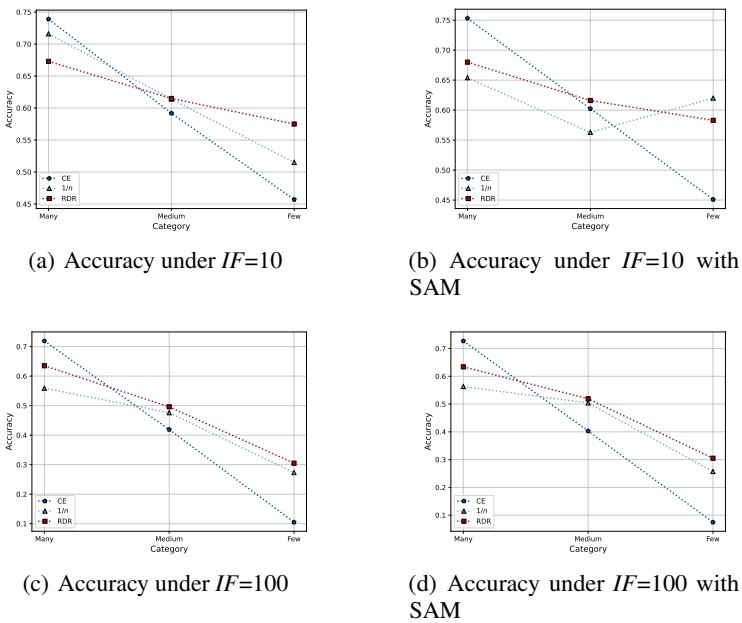

(a) Accuracy under *IF*=10

(b) Accuracy under *IF*=10 with SAM

(c) Accuracy under *IF*=100

(d) Accuracy under *IF*=100 with SAM

Figure 8: Visualization of top-1 accuracy (↑) across different categories on CIFAR-100-LT, under three different methods: CE, Inverse Frequency ($1/n$) and RDR. Experiments conducted under *IF*=10 (Plot (a) and Plot (b)) and 100 (Plot (c) and Plot (d)).

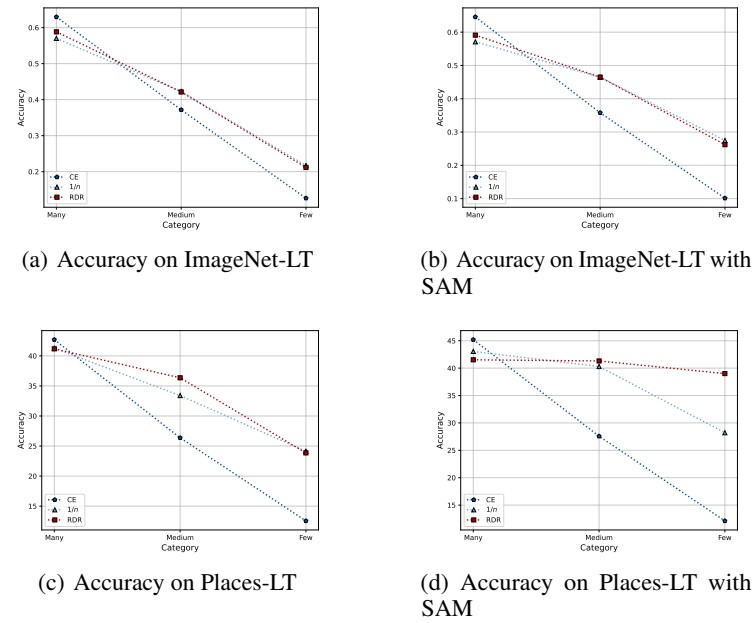

(a) Accuracy on ImageNet-LT

(b) Accuracy on ImageNet-LT with SAM

(c) Accuracy on Places-LT

(d) Accuracy on Places-LT with SAM

Figure 9: Visualization of top-1 accuracy (↑) across different categories on ImageNet-LT (Plot (a) and Plot (b)) and Places-LT (Plot (c) and Plot (d)), under three different methods: CE, Inverse Frequency ($1/n$) and RDR. Plot (b) and Plot (d) are integrated with SAM while Plot (a) and Plot (c) are not.

# E   More Discussions about Limitations

While our approach demonstrates promising results, there are potential challenges that warrant further attention. In particular, as the number of classes increases to a very large scale, especially in certain tasks such as face recognition, retail product recommendation, or landmark detection, there could be concerns regarding computational efficiency. It is important to consider lightweight techniques to ensure that scalability does not compromise practical applicability. Additionally, in addressing imbalanced learning, care must be taken to avoid excessive rebalancing toward minority groups, as this could unintentionally affect the learning performance of the majority class, which would not align with the broader goals of fairness. Rebalancing should be conducted within a reasonable framework, mindful of avoiding misuse or overcompensation that could arise from improper manipulation by any group. Ensuring fairness for all remains a critical consideration.

