# OpenReview forum: "Revive Re-weighting in Imbalanced Learning by Density Ratio Estimation"
_NeurIPS.cc/2024/Conference — NeurIPS 2024 poster_

### Official Review · Reviewer_24eD · 2024-06-29

**Soundness:** 3
**Presentation:** 2
**Contribution:** 3
**Rating:** 6
**Confidence:** 4

**Summary:**

This paper presents a dynamic re-weighting method for imbalanced learning. The author defines the ratio of the balanced data set distribution to the training set distribution, and tries to estimate it with an iterative update method. The effectiveness of this method is proved by experiments.

**Strengths:**

1.	This paper points out a problem with distribution differences, which leads to the potential missing feature patterns in general re-weighting methods.
2.	This paper proposes a new method, which approximates the ratio of the balanced data set distribution to the training set distribution using methods of density ratio estimation. As far as I know, a dynamic re-weighting strategy is novel in this field.
3.	The experimental introduction of this paper is clear, and extensive experiments have been carried out, which validates the effectiveness of the proposed method.

**Weaknesses:**

1.	The formula derivation in Sec. 3.3 can be more detailed. It is suggested to explain how formula (7) is obtained in the appendix.
2.	The introduction may have overlooked some key articles. For example, the article mentions Wang et al. 's article at the end of Sec.3.3, but does not discuss this paper in the introduction section.
3.	Does the new method enjoy the same theoretical boundaries as the general reweighting method? It is recommended to provide more analysis.
4.	Besides, there are some typos in the details:
  - In the experimental section, 'class[390,385] 'may be a typo.
  - In table 3, the interpretation of Tr_{Few} is supposed to be there.

**Questions:**

Please refer to Weaknesses.

Besides, I am also curious about some experimental details. Do the authors use other techniques such as RandAug or mixup?

**Limitations:**

No. Although the authors say they clarify the limitations in Sec.3.4, I find they mainly highlight the efficiency of the proposed method. More discussion about limitations and potential negative societal impact is recommended.

---

> ### Author Rebuttal · Authors · 2024-08-07
>
> > The formula derivation in Sec. 3.3 can be more detailed. It is suggested to explain how formula (7) is obtained in the appendix.
>
> Thank you for the suggestion. We will include detailed derivations of Eq. (7) in the appendix of revision, which we simply summarize the deduction as follows for clarity:
>
> Back to Eq. (5), replace the expectations over $P _ {bal}$ and $P$ by $\boldsymbol{\Phi} _ {P}$ and $F _ {P _ {bal}}$, respectively. For each class $i$, We can obtain
>
> $\widehat{r _ {i}}=\widehat{\mathrm{MM}(r)}$, where $\widehat{\mathrm{MM}(r)}=
> \frac{1}{n _ {i}^{2}}{r _ i}^\top{\boldsymbol{\Phi} _ {P}^{i}}^{\top} {\boldsymbol{\Phi} _ {P}^{i}}{r _ i}-\frac{2}{n _ {i}}r _ {i}^\top{\boldsymbol{\Phi} _ {P}^{i}}^\top F _ {i}$
>
> Then, taking the derivative of $\widehat{\mathrm{MM}(r)}$ with respect to $r$ and setting it to zero, we can obtain the estimation of density ratio in imbalanced learning as follows
>
> $\frac{2}{n _ {i}^{2}} {\boldsymbol{\Phi} _ {P}^{i}}^{\top} {\boldsymbol{\Phi} _ {P}^{i}} r _ {i}-\frac{2}{n _ {i}}{\boldsymbol{\Phi} _ {P}^{i}}^{\top}F _ {i}=0$
>
> Solving equation above with respect to $r _ {i}$, we can obtain the solution as
>
> $\widehat{r _ {i}}=n _ {i}\left({\boldsymbol{\Phi} _ {P}^{i}}^{\top} \boldsymbol{\Phi} _ {P}^{i}\right)^{-1} {\boldsymbol{\Phi} _ {P}^{i}}^{\top} F _ {i}$
>
> > The introduction may have overlooked some key articles. For example, the article mentions Wang et al. 's article at the end of Sec.3.3, but does not discuss this paper in the introduction section.
>
> We will take the reviewers' advice to carefully revise the second paragraph of the Introduction (lines 32-34), specially including a discussion of the work by Wang et al. and other related studies. We present the corresponding revision as follows for your reference.
>
> However, such subsequent improvements can alleviate but still cannot effectively push that forward. *Wang et al. [2023] obtains a fine-grained generalization bound for re-weighting in imbalanced learning through the data-dependent contraction technique.* Limited research has focused on the intrinsic limitations...
>
> > Does the new method enjoy the same theoretical boundaries as the general reweighting method? It is recommended to provide more analysis.
>
> Thank you for your suggestion. To address the reviewer's concern, we present a generalization bound sketch for the re-weighting-based methods and provide the possible insights with the empirical verifiction in the following. Note that, about its formal version with the complete hypothesis and the deduction procedure, we will update to the manuscript in the final.
>
> Given the model $m\in M$ and a reweighting loss function $L_{RW}$, for any $\delta \in(0,1)$, with probability at least $1-\delta$ over the training set $\mathcal{S}$, the following generalization bound holds for the risk on the balanced distribution
>
> $\mathcal{R} _ {\text {bal}}^{L}(m) \precsim \Phi\left(L _ {RW}, \delta\right)+\frac{\mathfrak{S} _ {\mathcal{S}}(M)}{d \pi _ {1}} \sum _ {y=1}^{d} w _ y \sqrt{\pi _ {y}}\left[1-\operatorname{softmax}\left(B _ {y}(m)\right)\right]$
>
> where $\Phi\left(L _ {RW}, \delta\right)$ is positively correlated with the empirical reweighting risk of training set. $\mathfrak{C} _ {\mathcal{S}}(M)$ denotes the empirical complexity of the function set $M$. $B _ {y}(f)$ denotes the minimal prediction on the ground-truth class $y$ in the training set. $w _ y$ refers to the weight of class $y$ of the reweighting loss $L _ {RW}$. The formal theory and the proof will presented in the revision.
>
> We can get some insights from the above generalization bound:
>
> 1. **Why reweighting is necessary**: $w _ y$ helps to rebalancing the imbalanced term $\sqrt{\pi_{y}}\left[1-\operatorname{softmax}\left(B _ {y}(m)\right)\right]$ to get a sharper bound.
> 2. **Why dynamic reweighting is necessary**: The term $B _ {y}(m)$ changes dynamically with model training. Therefore, we need a $w_y$ that can adapt dynamically to the changes of $B _ {y}(m)$.
> 3. **Why RDR works**: From Figure 1 in **the attached PDF**, we observe that the dynamic trend of the RDR weight aligns well with $\sqrt{\pi _ {y}}\left[1-\operatorname{softmax}\left(B _ {y}(m)\right)\right]$, denoted as $B _ y^{\prime}$. This demonstrates that our RDR can adapt to the dynamic changes in $B _ y^{\prime}$, maintaining a sharp bound during dynamic training.
>
> > Besides, there are some typos in the details:
> > In the experimental section, 'class[390,385] 'may be a typo.
> > In table 3, the interpretation of Tr_{Few} is supposed to be there.
> > Do the authors use other techniques such as RandAug or mixup?
>
> Thank you for pointing it out. We have carefully proofread the whole manuscript, corrected typos, and added a table about necessary explanations of some notations or terms (please refer to the notation table in the **attached PDF file**). $\overline{\textit{Tr}}_{Few}$ denotes average trace of Hessian matrix over Few classes. These changes will be included in the revision. We used RandAug in our experiments but did not include mixup, for fair competition.
>
> >  More discussion about limitations and potential negative societal impact is recommended.
>
> Thanks for your suggestion. We will enrich the discussion about limitation and social impact in the revision. Specially, we will highlight the potential computational complexity issue when scaling up to a super large number of classes (e.g., in face recognition, retail product recommendation and landmark detections), which should pay additional attention to some corresponding lightweight techniques. Regarding the potential negative societal impact, we should clearly recognize that overly rebalancing to minority groups during training may bring the desctructive effect on the learning of majority groups, which is also not the desirable goal of imbalanced learning. All rebalancing techniques should build on top of a proper range for fairness and we should avoid some improper abuse by the malicious minority groups.

---

> ### Author Response · Authors · 2024-08-10
>
> Dear Reviewer 24eD,
>
> We genuinely appreciate your detailed feedback and the insightful comments you've provided on our manuscript.
>
> We have provided more clarifications and explanations as suggested. Additionally, we have discussed limitations and potential negative societal impact.
>
> Please let us know if anything is unclear. We truly appreciate this opportunity to improve our work and shall be grateful for any feedback you could give to us.
>
>
> Best Regards,
>
> The authors of Submission 4464

---

> ### Comment · Reviewer_24eD · 2024-08-13
>
> Thanks for your rebuttal! My concerns have been clarified. Hence, I will increase my rating accordingly.

---

> > ### Author Response · Authors · 2024-08-13
> >
> > Dear Reviewer 24eD,
> >
> > We greatly appreciate your time and effort in reviewing our responses and contributing to the enhancement of this paper. We will carefully follow your suggestions to incorporate all the points of our rebuttal in the revised version.
> >
> > Best,
> >
> > The authors of Submission 4464

---

### Official Review · Reviewer_MG5X · 2024-06-30

**Soundness:** 4
**Presentation:** 2
**Contribution:** 2
**Rating:** 6
**Confidence:** 3

**Summary:**

The paper introduces a novel approach called Re-weighting with Density Ratio (RDR) to address the challenges posed by imbalanced data distributions in machine learning.  The RDR approach aims to mitigate overfitting on majority classes and enhance adaptability across diverse datasets by continuously updating the weights in response to observed shifts in class density.  Extensive experiments on various large-scale, long-tailed datasets demonstrate that the RDR method significantly improves the model's generalization capabilities, particularly under severely imbalanced conditions.  The analysis of the weight changes during training reveals that the method increasingly focuses on minority classes as training progresses, initially learning common features across all categories and then targeting learning towards minority samples to enhance generalizability.  The paper also provides an ablation study to further validate the effectiveness of the proposed approach.

**Strengths:**

1. The paper introduces a novel approach called Re-weighting with Density Ratio (RDR) to address the challenges posed by imbalanced data distributions in machine learning.

2. Extensive experiments on various large-scale, long-tailed datasets demonstrate that the RDR method significantly improves the model's generalization capabilities, particularly under severely imbalanced conditions.

3. The analysis of the weight changes during training reveals that the method increasingly focuses on minority classes as training progresses, initially learning common features across all categories and then targeting learning towards minority samples to enhance generalizability.

4. The paper provides an ablation study to further validate the effectiveness of the proposed approach.

5. The results show that RDR generally outperforms other methods, including Inverse Frequency (1/n) and SAM variants, in both the Many and Few classes, indicating that RDR can efficiently address the overfitting issues for Few classes.

**Weaknesses:**

- The paper does not provide a detailed theoretical analysis or justification for the proposed Re-weighting with Density Ratio (RDR) method, beyond the intuition that it can mitigate overfitting on majority classes and enhance adaptability across diverse datasets.

- I am interested in how RDR might perform in the presence of extreme imbalance, noisy data, or other challenging scenarios. The current experiment is well-established but dataset itself is relatively simple.

- The paper discusses reweighting/non-reweighting for classification problems. I suggest the authors also briefly discuss reweighting methods in imbalanced regression problems, e.g., VIR [1] for reweighting problems and ConR [2] for non-reweighting problems.

[1] Variational Imbalanced Regression: Fair Uncertainty Quantification via Probabilistic Smoothing, NeurIPS 2023

[2] ConR: Contrastive Regularizer for Deep Imbalanced Regression, ICLR 2024

**Summary** I think the theoretical analysis or at least insights is needed for acceptance, so my suggest score is 5, as the experiment part is excellent in this paper.

**Questions:**

see above

**Limitations:**

authors discussed in sec 3.4

---

> ### Author Rebuttal · Authors · 2024-08-07
>
> > The paper does not provide a detailed theoretical analysis or justification for the proposed Re-weighting with Density Ratio (RDR) method, beyond the intuition that it can mitigate overfitting on majority classes and enhance adaptability across diverse datasets.
>
> Thank you for your suggestion. To address the reviewer's concern, we present a generalization bound sketch for the re-weighting-based methods and provide the possible insights with the empirical verifiction in the following. Note that, about its formal version with the complete hypothesis and the deduction procedure, we will update to the manuscript in the final.
>
> Given the model $m\in M$ and a reweighting loss function $L _ {RW}$, for any $\delta \in(0,1)$, with probability at least $1-\delta$ over the training set $\mathcal{S}$, the following generalization bound holds for the risk on the balanced distribution
>
> $\mathcal{R} _ {\text {bal }}^{L}(m) \precsim \Phi\left(L _ {RW}, \delta\right)+\frac{\mathfrak{S} _ {\mathcal{S}}(M)}{d \pi_{1}} \sum _ {y=1}^{d} w _ y \sqrt{\pi _ {y}}\left[1-\operatorname{softmax}\left(B _ {y}(m)\right)\right]$
>
> where $\Phi\left(L _ {RW}, \delta\right)$ is positively correlated with the empirical reweighting risk of training set. $\mathfrak{C} _ {\mathcal{S}}(M)$ denotes the empirical complexity of the function set $M$. $B _ {y}(f)$ denotes the minimal prediction on the ground-truth class $y$ in the training set. $w_y$ refers to the weight of class $y$ of the reweighting loss $L _ {RW}$. The formal theory and the proof will presented in the revision.
>
> We can get some insights from the above generalization bound:
>
> 1. **Why reweighting is necessary**: $w _ y$ helps to rebalancing the imbalanced term $\sqrt{\pi _ {y}}\left[1-\operatorname{softmax}\left(B _ {y}(m)\right)\right]$ to get a sharper bound.
> 2. **Why dynamic reweighting is necessary**: The term $B _ {y}(m)$ changes dynamically with model training. Therefore, we need a $w _ y$ that can adapt dynamically to the changes of $B _ {y}(m)$.
> 3. **Why RDR works**: From Figure 1 in **the attached PDF**, we observe that the dynamic trend of the RDR weight aligns well with $\sqrt{\pi _ {y}}\left[1-\operatorname{softmax}\left(B _ {y}(m)\right)\right]$, denoted as $B _ y^{\prime}$. This demonstrates that our RDR can adapt to the dynamic changes in $B _ y^{\prime}$, maintaining a sharp bound during dynamic training.
>
> > I am interested in how RDR might perform in the presence of extreme imbalance, noisy data, or other challenging scenarios. The current experiment is well-established but dataset itself is relatively simple.
>
> Thank you for the suggestion. In the following, we conduct more experiments. The results are shown in Table 2 and Table 3 in the **attached PDF**.
> - We include the results of two new datasets: CIFAR-10-LT-NL and CIFAR-100-LT-NL, with both class imbalance and label noise. We can see that on more complex datasets, our method achieves consistent and significant improvements.
> - Additionally, note that the imbalance factors constructed on ImageNet-LT and Places-LT in the submission are 256 and 996, respectively. These datasets are indeed extremely imbalanced. We will explicitly indicate the imbalance factors of these two datasets in the experiment tables in the revision. Furthermore, we conduct experiments on the more imbalanced CIFAR-10-LT and CIFAR-100-LT datasets, specifically with imbalance factors of 200 and 500. Our method consistently achieves significant improvements.
>
>
> > The paper discusses reweighting/non-reweighting for classification problems. I suggest the authors also briefly discuss reweighting methods in imbalanced regression problems, e.g., VIR [1] for reweighting problems and ConR [2] for non-reweighting problems.
> [1] Variational Imbalanced Regression: Fair Uncertainty Quantification via Probabilistic Smoothing, NeurIPS 2023
> [2] ConR: Contrastive Regularizer for Deep Imbalanced Regression, ICLR 2024
>
> Thank you for the valuable suggestion and the recommendation. We will include a brief discussion on imbalanced regression problems in the revision as follows:
>
> We shall note that the primary focus of this study is on imbalanced classification. Beyond imbalanced classification that has discretized label space, an noteworthy area, imbalanced regression that has a continuous label space is also very common in real applications [a,b]. In this direction, the empirical label distribution often does not accurately reflect the true label density in regression tasks, which limits the effectiveness of traditional reweighting techniques [a,c]. Label Distribution Smoothing (LDS) [a] and Variational Imbalanced Regression (VIR) [c] propose using kernel smoothing and other techniques to estimate an accurate label density distribution. Ranking Similarity (Ranksim) [b] leverages local and global dependencies by encouraging the correspondence between the similarity order of labels and features. Balanced Mean Squared Error (Balanced MSE)[d] extends the concept of Balanced Softmax[e] to regression tasks to achieve a balanced predictive distribution. Contrastive Regularizer (ConR)[f] improves contrastive learning techniques to translate label similarities into the feature space. Considering the different rebalancing paradigms compared with that in imbalanced classification, and the limited space, we will leave the potential extension of our RDR to this area in the future explorations.
>
> [a] Delving into Deep Imbalanced Regression. ICML 2021.
>
> [b] RankSim: Ranking Similarity Regularization for Deep Imbalanced Regression. ICML 2022.
>
> [c] Variational Imbalanced Regression: Fair Uncertainty Quantification via Probabilistic Smoothing. NeurIPS 2023.
>
> [d] Balanced MSE for Imbalanced Visual Regression. CVPR 2022.
>
> [e] Balanced Meta-Softmax for Long-Tailed Visual Recognition. NeurIPS 2020.
>
> [f] ConR: Contrastive Regularizer for Deep Imbalanced Regression. ICLR 2024.

---

> > ### Comment · Reviewer_MG5X · 2024-08-09
> > **reviewer update**
> >
> > I would like to thank the authors for their detailed response. I will take all the reviews and responses into consideration.

---

> > ### Comment · Reviewer_MG5X · 2024-08-11
> > **score update**
> >
> > It seems that the other reviewers have not responded yet. After reviewing the authors' responses to the other reviewers, I have decided to raise my score. However, I **cannot** promise that the authors' responses address the concerns of the other reviewers, so the authors **should not use my improved score as a reference or evidence**.
> >
> > In summary, I appreciate the authors' responses to all of us. Specifically, I feel that the authors have addressed my questions, so I have decided to increase my score.

---

> > > ### Author Response · Authors · 2024-08-11
> > >
> > > Dear Reviewer MG5X,
> > >
> > > We sincerely appreciate you taking the time to review our responses and contributing to improve this paper. We will carefully follow reviewer's advice to incorporate all the addressed points with additional experiments in the updated version.
> > >
> > > We promise not to use your improved score as evidence to persuade other reviewers, but focus on truly addressing their concerns.
> > >
> > > Thank you once again for your dedicated and valuable contribution in reviewing our paper!
> > >
> > > Best,
> > >
> > > The authors of Submission 4464

---

### Official Review · Reviewer_xrua · 2024-07-08

**Soundness:** 3
**Presentation:** 1
**Contribution:** 3
**Rating:** 6
**Confidence:** 3

**Summary:**

The paper presents a weighting strategy in order to handle class imbalance. Contrary to existing method, they propose to adapt the weight throughout the training procedure.

Their method estimates the discrepancy between the sample distribution and the balanced sample distribution for parameterization w and updates the estimate through the training.

The authors use two resnet architectures to evaluate their contribution on multiple datasets. They also compare to other baselines and show significant gain.

**Strengths:**

* The paper develops a novel approach for handling class imbalance.
* The methodology is derived theoretically from the problem formulation
* The authors propose an analysis of the complexity of the method and empirically evaluate the training time.
* The methodology is evaluated on multiple datasets and compared to multiple baselines.

**Weaknesses:**

* The paper is sometimes difficult to read:
  * Row 125, the authors refer to the distribution of training set, which get parameterized by w. Thus, my understanding is that the authors refer to the distribution of the training set "captured by the model".
  * row 134, P_bal = pi P.. P_bal is the distribution of y in the balanced case ? But should therefore be 1/number classes... and P, should just be the class proportion and we should have P = pi P_bal ?
  * LDAM and LA terms are not defined at first. First definition of LA is at row 208
  * row 212 "trategies" => strategies

**Questions:**

* Could you clarify the term, P, etc. You often refer to it as "real world data distribution", but the distribution of x does not depends on any other classes (imbalanced or not) ?

**Limitations:**

The paper would benefit from a clarification in notification (what is the true data distribution, what is the feature distribution, what is an estimate of what quantity, etc.). I believe the contribution is novel and worth it.

---

> ### Author Rebuttal · Authors · 2024-08-07
>
> > Row 125, the authors refer to the distribution of training set, which get parameterized by w. Thus, my understanding is that the authors refer to the distribution of the training set "captured by the model".
>
> Yes, you are right. We use the distribution parameterized by $w$ to represent "the distribution captured by the model". We will enrich its description for clarity in the revision.
>
> > row 134, P_bal = pi P.. P_bal is the distribution of y in the balanced case ? But should therefore be 1/number classes... and P, should just be the class proportion and we should have P = pi P_bal ?
>
> Thank you for pointing out this oversight: the term  $\pi_i$  in the equation should indeed be $\frac{1}{\pi_i}$. We would like to clarify here that this mistake is limited to this equation and does not affect the correctness of our subsequent derivations. We greatly appreciate the reviewer’s careful review and will make the correction.
>
> > LDAM and LA terms are not defined at first. First definition of LA is at row 208
>
> > row 212 "trategies" => strategies
>
> LDAM and LA respectively refer to Label-Distribution-Aware margin loss and Logit Adjustment, two prevalent long-tailed learning methods. We will include their full names at the first mention. For the typo, we have carefully proofread the whole manuscript again and corrected all possible mistakes. These changes will be included in the revision.
>
> > Could you clarify the term, P, etc. You often refer to it as "real world data distribution", but the distribution of x does not depends on any other classes (imbalanced or not) ?
>
> Sorry for the possible confusion. Here is a clarification of $P$:
>
> $P$ represents the real world data distribution, or more directly, the training set distribution. Specifically, $P(x,y)$ is the joint distribution of the training data (x, y), and the training set $D$ is obtained by IID sampling from $P$. $P(x)$ and $P(y)$ represent the marginal distributions of the training data $x$ and $y$, respectively. $P(x|y)$ and $P(y|x)$ represent the corresponding conditional distributions. For other distribution notations parameterized by $w$, they mean the corresponding distributions captured by the model, as the reivewer's understanding.
>
> We also place a notation table for clarity in the **attached PDF file**, which will be placed into the manuscript if possible (or in the appendix if not). If you have any further questions or need any clarifications, please let us know. We will carefully improve the manuscript.

---

> ### Author Response · Authors · 2024-08-10
>
> Dear Reviewer xrua,
>
> We sincerely appreciate your constructive feedback and positive evaluation of our submission.
>
> We have provided more clarifications and explanations, made necessary corrections, and improved definitions as suggested.
>
> Please let us know if anything is unclear. We truly appreciate this opportunity to improve our work and shall be most grateful for any feedback you could give to us.
>
> Best Regards,
>
> The authors of Submission 4464

---

> > ### Comment · Reviewer_xrua · 2024-08-13
> > **Answer**
> >
> > Dear authors,
> >
> > I thank you for your answers. After considering your propositions and mostly the enhanced readability (e.g. symbol table), I have decided to raise my note.
> >
> > Regards

---

> > > ### Author Response · Authors · 2024-08-13
> > >
> > > Dear Reviewer xrua,
> > >
> > > We sincerely appreciate you taking the time to review our responses and contributing to improve this paper. We will carefully follow your advice to incorporate all the points of our rebuttal in the updated version.
> > >
> > > Best,
> > >
> > > The authors of Submission 4464

---

### Author Rebuttal · Authors · 2024-08-07

We thank reviewers for your valuable feedback, and appreciate the great efforts made by all reviewers, ACs, SACs and PCs.

Please refer to our detailed responses to each reviewer, where we addressed each question and concern point by point. In the **attached PDF**, we have included a notation summary table for improved readability and clarification, additional tables for experimental results, along with the empirical validation of the theoretical insights presented in the rebuttal.

We appreciate all reviewers’ time again. We are looking forward to your reply!

---

### Decision · Program_Chairs · 2024-09-25

**Decision:**

Accept (poster)

**Comment:**

The authors address the critical and challenging problem of class imbalance in machine learning. They propose a method that adaptively weighs the importance of classes throughout training. The main idea is to use an estimate of the density ratios (existing vs. balanced) to re-weight the samples in the loss. The advantage of the method is demonstrated using several real datasets with imbalance ratios ranging from 10 to 256. Overall, the reviewers are positive about the paper, with the main criticism being the presentation. This work provides an interesting solution to the problem that seems to work well. For these reasons, I recommend acceptance of this paper, and I highly encourage the authors to implement the reviewers' suggestions and work towards improving the writing and presentation.